# Tillage Promotes the Migration and Coexistence of Bacteria Communities from an Agro-Pastoral Ecotone of Tibet

**DOI:** 10.3390/microorganisms10061206

**Published:** 2022-06-13

**Authors:** Yuhong Zhao, Mingtao Wang, Yuyi Yang, Peng Shang, Weihong Zhang

**Affiliations:** 1Department of Animal Science, Tibet Agricultural and Animal Husbandry University, Nyingchi 860000, China; zhaoyh@xza.edu.cn (Y.Z.); wmt060708@126.com (M.W.); nemoshpmh@126.com (P.S.); 2Key Laboratory of Aquatic Botany and Watershed Ecology, Wuhan Botanical Garden, Chinese Academy of Sciences, Wuhan 430074, China; yangyy@wbgcas.cn; 3University of Chinese Academy of Sciences, Beijing 100049, China

**Keywords:** tillage, soil nutrients, bacterial community, coexistence, agro-pastoral region

## Abstract

In the Tibetan agro-pastoral ecotone, which has an altitude of 4000 m above sea level, small-scale cropland tillage has been exploited on the grassland surrounding the houses of farmers and herdsmen. However, knowledge of the effects of land change from grassland to cropland on soil nutrients and microbial communities is poor. Here, we investigated the structure and assembly mechanism of bacterial communities in cropland (tillage) and grassland (non-tillage) from an agro-pastoral ecotone of Tibet. Results indicated that soil nutrients and composition of bacterial communities changed dramatically in the process of land-use change from grassland to cropland. The pH value and the content of total nitrogen, organic material, total potassium, and total phosphorus in cropland soil were well above those in grassland soil, whereas the soil bulk density and ammonia nitrogen content in grassland soil were higher than those in cropland soil. *Proteobacteria* (30.5%) and *Acidobacteria* (21.7%) were the key components in cropland soil, whereas *Proteobacteria* (31.5%) and *Actinobacteria* (27.7%) were the main components in grassland soils. Tillage promotes uniformity of bacterial communities in cropland soils. In particular, the higher migration rate may increase the coexistence patterns of the bacterial community in cropland soils. These results also suggest that the tillage promotes the migration and coexistence of bacterial communities in the grassland soil of an agro-pastoral ecotone. In addition, the stochastic process was the dominant assembly pattern of the bacterial community in cropland, whereas, in grassland soil, the community assembly was more deterministic. These findings provide new insights into the changes in soil nutrients and microbial communities during the conversion of grassland to cropland in the agro-pastoral ecotone.

## 1. Introduction

The agro-pastoral ecotone is the transitional buffer area of the agriculture and animal husbandry ecosystem, and has great productive potential [1]. The ecological environment and sustainable economic production problems in the agro-pastoral ecotone have attracted extensive attention from researchers with the intensification of human activities and global climate change [1,2,3]. The content of the research mainly involves tillage systems, land degradation, land restoration, and other aspects [1,2,4]. However, most scholars pay attention to the agro-pastoral ecotone in northern China but ignore the Tibet-Yunnan-Sichuan agro-pastoral ecotone in southern China [1,2,3,4].

The agro-pastoral ecotone of Tibet is one of the most sensitive and vulnerable regions on the plateau, and plays a crucial role in the ecological environment safety and national economic development of locals [5]. Under the influence of long and unique traditional cultural habits and the natural environment, a comprehensive industrial structure of planting and animal husbandry has been formed. Grassland and cropland are the main land-use types in the agro-pastoral interlaced area in southeast Tibet [6]. Grassland protects biodiversity and water and soil conservation of the Qinghai-Tibet Plateau ecosystem [7]. However, small-scale cropland tillage has been exploited on grassland surrounding the houses of farmers and herdsmen with the continuous development of the economy and the increase in the population. As a result, some of the original grasslands in the valley have become cropland. It is reported that land use changed the soil nutrient cycling and microbial community on the Qinghai-Tibet Plateau [8]. It is worth noting that, after the grassland in the agro-pastoral ecotone of Tibet was transformed into farmland, not only was the land use changed, but the tillage system was also changed. Grassland in Tibet represents a no-tillage system, whereas the main tillage system of cropland is the ox-pulled plow. Therefore, understanding the response of soil nutrients and the composition of microbial communities to the change in the tillage system is of great significance.

Microbial community composition is an important factor affecting soil function [9]. In recent studies, some scattered studies have been conducted on soil microorganisms of different land uses in the Qinghai-Tibet Plateau. Ref. [6] found that soil pH was the main soil attribute that affected the bacterial community composition of cropland, forest, and grassland on the Qinghai-Tibet Plateau. Ref. [8] studied the composition and function of microbial communities following the land-use change from forest to grassland in the Eastern Tibetan Plateau and found distinct physicochemical and microbial communities. The composition of the microbial community was mainly regulated by total nitrogen (TN) and pH [8]. The cropland in the agro-pastoral ecotone of Tibet was established on grassland soil. However, the consequences of tillage system change on soil nutrients and microbial communities during this process are poorly understood. For those reasons, the composition and assembly processes of bacterial communities in cropland and grassland in an agro-pastoral ecotone of Tibet were investigated, and the relationship between soil nutrient changes and bacterial community differences was analyzed.

## 2. Materials and Methods

### 2.1. Overview of Experimental Field and Soil Sample Collection

To study the variation in soil physicochemical properties and bacterial communities in cropland and grassland from the agro-pastoral ecotone, Bangda town (E 97.290556°; N 30.174444°), which is located in the valleys of the southern part of the Qinghai-Tibet Plateau, was taken as an example. This is a typical agro-pastoral ecotone in the Qinghai-Tibet Plateau, having an altitude of 4000 m above sea level and a plateau temperate semi-arid monsoon climate. The annual frost-free period is 161.7 days, and the annual precipitation is 233.3 mm. In this study, the surficial soil (0–10 cm) was collected with a stainless-steel core sampler and three sub-samples were mixed as one sample. In the agro-pastoral ecotone of Tibet, the traditional ox-pulled plow is used as the main tillage system. Given the mix of agricultural and pastoral areas in Tibet, part of the cropland soil is used for planting wheat and highland barley crops (wheatland, WL), and part is was used for cultivated pasture (CP). Part of the grassland is fenced to ban grazing (enclosure grassland, EG) and part is used for grazing (natural grassland, NG). Six cropland soil samples were sampled from two types of cropland (three WL samples and three CP samples) and six grassland soil samples were sampled from two types of grassland (three NG samples and three EG samples). Finally, a total of 12 soil samples were collected and stored at −80 °C before analysis.

### 2.2. Soil Nutrients and Physicochemical Properties

The soil metrics included soil water content (SWC), pH, soil bulk density (SBD), total nitrogen (TN), total phosphorus (TP), total potassium (TK), ammonia nitrogen (NH_4_-N), nitrate-nitrogen (NO_3_-N), available phosphorus (AP), available potassium (AK), and organic matter (OM). These soil physicochemical and nutrient variables were quantified according to methods described by Qiu et al. [10].

### 2.3. 16S rRNA Sequencing

Genomic DNA of the bacterial community from each soil sample was extracted using the DNeasy PowerSoil Kit (QIAGEN, Inc., Venlo, The Netherlands) according to the manufacturer’s instructions. When DNA extraction was completed, the forward primer 515F (5-GTGCCAGCMGCCGCGGTAA-3) and the reverse primer 907R (5′-CCGTCAATTCMTTTRAGTTT-3′) were used for PCR amplification of the bacterial 16S rRNA gene V4-V5 region. Then, the PCR amplicons were purified and quantified via Vazyme VAHTSTM DNA Clean Beads and Vazyme VAHTSTM DNA Clean Beads, respectively. Illumina MiSeq platform with MiSeq Reagent Kit v3 was performed to bacterial community sequencing. After the sequencing, low-quality components and chimeric sequences were removed from the raw fastq data through sequence quality control to obtain clear reads. The remaining high-quality sequences were clustered into operational taxonomic units (OTUs) with more than 97% sequence consistency, and their taxonomic information was annotated by the VSEARCH software (version 2.3.4) analysis process [11].

### 2.4. Data Analyses

The shared patterns of bacterial communities within cropland and grassland, and between cropland and grassland, were described utilizing the Venn diagram. OTUs having a detection rate of 100% in cropland or grassland were thought to be their “persistent bacteria”. Correlations of alpha diversity and composition of bacterial communities in cropland and grassland with their corresponding soil nutrients and physicochemical factors were tested via Mantel tests. The normalized stochasticity ratio (NST) was calculated using the “NST” package of R to determine the relative contributions of deterministic and stochastic processes to bacterial community assembly in cropland and grassland. Furthermore, a neutral community model was used to determine the contribution of stochastic processes to microbial community assembly. Based on Spearman’s correlation (|r| > 0.7, *p* < 0.05), co-occurrence networks and topology properties of bacterial communities in cropland and grassland were described using the Gephi, respectively.

## 3. Results

### 3.1. Soils Physicochemical Properties and Nutrients in Cropland and Grassland

Physicochemical and nutrients were investigated in the soil of cropland and grassland. The pH in cropland was higher than that of grassland, and there was no significant difference in soil water content. However, the soil bulk density of cropland soil was lower than that of grassland soil (Appendix A). In addition, there were also significant differences in nutrients between grassland and cropland (Figure 1). The content of TN, OM, TK, and TP in cropland soils was significantly greater than that in grassland soil, whereas only the NH_4_-N content in grassland soils was well above that in cropland. Furthermore, there was no significant difference between the content of NO_3_-N, AK, and AP in cropland soils and grassland soils.

### 3.2. Diversity and Composition of the Bacterial Community in Cropland and Grassland Soils

The richness and diversity of the bacterial communities in cropland and grassland soils were described by 16S rRNA gene sequencing, and high coverage (Good’s coverage of >97.7%) was found (Appendix A). The rarefaction curves of the observed OTU number (Appendix A), Shannon (Appendix A), and Chao1 (Appendix A) indicated that all soil samples were deep sequenced and could be used for the analysis of the bacterial community. The alpha diversity index (Chao1, Shannon, and Pielou’s evenness index, and observed species) of bacterial communities were evaluated, and no significant difference was shown in the alpha diversity index between cropland and grassland soils (Appendix A).

Figure 2 shows the main components of the bacterial community in cropland and grassland soils. *Proteobacteria* (average relative abundance was 30.5%) was the main bacteria in cropland, followed by *Acidobacteria* (21.7%), *Actinobacteria* (17.1%), *Chloroflexi* (13.6%), and *Gemmatimonadetes* (7.32%). *Proteobacteria* (31.5%) were also the dominant bacteria in grassland, followed by *Actinobacteria* (27.7%), *Acidobacteria* (18.2%), *Chloroflexi* (10.9%), *Gemmatimonadetes* (4.54%). The results of the independent sample *t*-test testify to the relative abundance of *Acidobacteria*, *Bacteroidetes*, *BRC1*, *Chloroflexi*, *Deinococcus-Thermus*, *Fibrobacteres*, *Gemmatimonadetes*, *Patescibacteria*, *Planctomycetes*, *Rokubacteria*, and *WS2* in cropland soils being significantly higher than that in grassland soils (Appendix A). However, the relative abundance of *Actinobacteria* and *Cyanobacteria* in grassland was more than that in cropland (Appendix A). This suggests there were distinct differences in the composition of bacterial communities between cropland and grassland.

### 3.3. Influence of Environmental Variables on Bacterial Communities in Cropland and Grassland Soils

The Mantel test showed that NH_4_-N (r = 0.70, *p* < 0.05) was significantly correlated with the alpha diversity of bacterial communities in grassland, whereas environmental factors in cropland did not have a significant correlation with their corresponding alpha diversity indices (Figure 3). This demonstrated that bacterial species diversity in grassland soils might be closely correlated with NH_4_-N content. In addition, the Mantel test showed that SWC (r = 0.83, *p* < 0.05), TN (r = 0.94, *p* < 0.05), NO_3_-N (r = 0.95, *p* < 0.05), AP (r = 0.95, *p* < 0.05), OM (r = 0.95, *p* < 0.05), TK (r = 0.96, *p* < 0.05), and AK (r = 0.95, *p* < 0.05) were significantly correlated with the composition of bacteria in grassland soils, whereas these environmental factors in cropland soils did not show a significant correlation with their corresponding composition of bacterial communities (Figure 3). These findings demonstrate that the composition of bacterial communities in grassland soils was more susceptible to environmental variables than in cropland soils. These results also prove that the composition of bacterial communities in grassland soils was more susceptible to environmental variables than alpha diversity, and different environmental variables have different effects on the composition and alpha diversity of bacterial communities in grassland.

### 3.4. Persistent Bacteria in Cropland and Grassland Soils

At the OTU levels, the persistent bacteria in cropland and grassland were further identified (Figure 4). The number of OTUs of persistent bacteria in cropland (518) was higher than that in grassland (123). This suggests that more bacterial species persist in cropland. The relative abundance of persistent bacteria in cropland was 42.54%, and was composed of OTUs belonging to 15 phyla. *Proteobacteria* (13.54%), *Acidobacteria* (9.89%), *Bacteroidetes* (8.53%), *Chloroflexi* (5.91%), and *Gemmatimonadetes* (2.85%) were the major component of persistent bacteria in cropland (Figure 4A). However, the relative abundance of persistent bacteria in grassland was 91.75%, which was composed of OTUs belonging to seven phyla. *Proteobacteria* (28.87%), *Actinobacteria* (23.77%), *Acidobacteria* (21.36%), *Chloroflexi* (9.23%), *Gemmatimonadetes* (5.99%), *Rokubacteria* (1.40%), and *Firmicutes* (1.14%) were the chief components of persistent bacteria in grassland soils (Figure 4B). Only 72 OTUs, having a relative abundance of 11.98%, were detected in all croplands and grasslands, and were affiliated with *Proteobacteria* (3.90%), *Acidobacteria* (2.95%), *Actinobacteria* (2.91%), *Chloroflexi* (1.10%), *Gemmatimonadetes* (0.73%), *Rokubacteria* (0.25%), and *Firmicutes* (0.13%) (Figure 4C).

### 3.5. Bacterial Communities Assembly in Cropland and Grassland

The NST results indicated that assembly of the bacterial community in cropland was more stochastic (NST > 50%), whereas assembly of the bacterial community in grassland soils was more deterministic (NST < 50%) (Figure 5A). The neutral model also showed that the stochastic process (R^2^ = 0.448) was the dominant assembly process of bacterial communities in cropland, whereas the deterministic process (R^2^ = −0.236) was the main assembly process of bacterial communities in grassland (Figure 5B,C). These findings indicate that bacterial communities in cropland soils and grassland soils have opposite assembly processes in the agro-pastoral ecotone of Tibet. The estimated migration rate (m) of bacterial communities in cropland soils (0.06) was higher than that in grassland soils (0.035) (Figure 5B,C), suggesting that grassland soils only had tiny bacterial communities that exchange with the surrounding environment compared to cropland soils.

### 3.6. Coexistence Patterns of Bacterial Communities in Cropland and Grassland

The number of nodes (528) and edges (12,518) of the co-occurrence network of bacterial communities in cropland soils was higher than that in grassland soils (122 nodes and 1153 edges, respectively) (Figure 6A,B). The percentage of positive correlation of networks in cropland soils (51.25%) was lower than that in grassland soils (56.72%). The nodes of the soil bacterial community network both in cropland and grassland were mainly *Proteobacteria*, *Acidobacteria*, and *Actinobacteria* (Figure 6C). The betweenness centrality of the network in cropland soils (447.5) was more than that in grassland soils (133.9) (Figure 6D). In addition, we investigated the key topological features of networks in cropland and grassland soils (Table 1). The result showed that the co-occurrence network of bacterial communities in cropland had a higher clustering coefficient (0.41), average connectivity (48.3), and module number (6), suggesting the network connectivity and complexity of the cropland bacterial community were stronger than those of the grassland bacterial community. However, the graph density (0.09) and average weighted degree (1.05) of the cropland network were lower than those of grassland (0.16 and 2.21, respectively). These results show that the relatively low numbers of OTUs associated with the network of bacterial communities in grassland tended to be closely potentially connected.

## 4. Discussion

### 4.1. Tillage Alters the Variation in Soil Physicochemical Properties and Nutrients in an Agro-Pastoral Ecotone

Soil physicochemical properties and nutrients are important tools for evaluating the soil health status and the basis for evaluating soil biological activity [12]. Previous studies confirmed that different land-uses lead to changes in soil physicochemical features and nutrients [12,13]. However, due to the diversification of land-use types, tillage types and methods, and their eco-geographical environment, the results concerning the impact of tillage on the impact of soil physicochemical features and nutrients, to date, have been rather ambiguous, and sometimes contradictory [14,15,16]. This study found that tillage promotes the variation in soil physicochemical features and nutrients in an agro-pastoral ecotone of Tibet.

Specifically, the soil bulk density decreased with the transformation of natural grasslands change into cropland, which was consistent with grasslands conversion to croplands in the Bashang area of Northern China [17]. Previous studies showed that the decrease in soil bulk density caused porosity to increase, and ensured more water in the soil of southwestern China [14], whereas the soil water content increased with the increase in the soil bulk density in the semi-arid area of China [16]. This study showed that the soil water content did not show a significant difference between the cropland and grassland. However, the soil water content of cropland changed greatly, which may be related to the different land-use types in cropland. Previous studies suggested that the pH between cropland and grassland showed a significant difference [18], and our study also reached a similar conclusion. Tillage was thought to increase soil erosion and reduce soil organic matter content [19]. However, this study found that soil OM content in cropland was higher than that in grassland. We speculate that the subsurface biomass of alpine grassland in Tibet was lower than that of crops, which leads to higher soil OM content in cropland than in grassland. Furthermore, it may also be due to less tillage occurring in the agro-pastoral ecotone of Tibet. Less tillage can effectively increase the contents of soil organic matter and other nutrients [20,21,22]. The contents of TN, TK, and TP in cropland exceeded those in grassland, which further confirmed that less tillage can effectively promote soil nutrients in the agro-pastoral ecotone of Tibet.

### 4.2. Tillage Shaping the Variation in the Composition of Bacterial Communities Rather Than Diversity

Bacterial communities are an important part of global biogeochemical processes, which had highly diverse at the taxonomy level [23]. This study found that the alpha diversity of bacterial communities in cropland was not distinct from in grassland. Previous studies found that the alpha diversity of the soil microbial community was not different in grassland and cropland in the agro-pastoral ecotone [24], which is consistent with this study. It can be concluded that tillage had little effect on the alpha diversity of the bacterial community in the grassland of agro-pastoral ecotone.

However, the composition and structure of bacterial communities in cropland and grassland soil showed significant differences. For example, the relative abundance of *Actinobacteria* and *Cyanobacteria* in grassland was higher than that in cropland. Xu et al. [24] also found that the soil microbial community composition rather than alpha diversity was affected by land use in the agro-pastoral ecotone, which was consistent with this study. Thus, the composition rather than the diversity of soil bacterial communities were affected by long-term tillage in an agro-pastoral ecotone. Furthermore, we found that more OTUs of bacterial communities were shared across all cropland than across all grassland. This indicates that tillage may promote bacterial community migration and colonization in cropland.

### 4.3. Proper Tillage May Promote Connectivity and Complexity of Soil Bacterial Network

Understanding the species interaction among soil bacterial communities is critical to predicting the function of the effect of biodiversity on ecosystems [25]. Network analysis provides unique and profound insights into the potential associations within bacterial communities and the rules of ecological assembly [26]. This study showed that the network connectivity and complexity of the cropland bacterial community were stronger than those of the grassland bacterial community. In contrast, grassland bacterial networks were relatively small, but they interact closely. This indicates that tillage increases the network connectivity and complexity of the cropland bacterial community but decreases its compactness.

However, some previous studies have found that, under high-intensity tillage, the network complexity of the cropland microbial community was greater compared with that of grassland [27,28], which was contrary to the results of this study. We speculate that this may be related to the simple and single farming method in the agro-pastoral ecotone of Tibet. In the agro-pastoral ecotone of Tibet, which has an altitude of 4000 m above sea level, the annual frost-free period is only 161.7 days, and only one crop can be planted with less tillage. These results also mean that proper tillage may promote connectivity and complexity of the soil bacterial network. Positive associations are often thought of as cooperation within the microbial community, whereas negative associations are often understood as competition relationships [26]. This study found that the cooperation within the bacterial communities in cropland was lower than that in grassland. It is widely believed that competitive exclusion makes communities more diverse by excluding closely related ecologically similar species [29]. This may be an important reason why the density of the cropland bacterial network was lower than that of the grassland bacterial network.

### 4.4. Bacterial Communities in Cropland and Grassland Had Opposite Assembly Processes

Determining the community assembly mechanisms that control microbial biogeographical patterns is a central goal of microbial ecology [26,30]. Deterministic and stochastic processes are two important mechanisms of microbial community assembly. The niche mechanism-based deterministic process suggests that microbial communities are influenced by environmental filtering (e.g., pH and nutrients) and various biological interactions (e.g., competition and cooperation) [29]. In this study, the composition of the grassland bacterial community may be susceptible to the regulation of SWC, TN, NO_3_-N, AP, OM, TK, and AK, which may be an important reason why the aggregation of the grassland bacterial community was dominated by a deterministic process. Similarly, the composition and diversity of cropland bacterial communities did not show a significant correlation with environmental factors, which may be an important reason for the opposite assembly process of cropland and grassland bacterial communities.

In addition, the stochastic process based on neutral theory suggests that microbial community assembly was often influenced by dispersal limitation, life-history traits (such as birth and death), and ecological drift (stochastic changes in biological abundance) [29]. Based on the neutral community model, we observed the bacterial communities in grassland had lower fitness values (R^2^) and estimated migration rate (m) compared to those in cropland. This suggests that the stochastic process plays a dominant role in shaping the assembly of cropland bacterial communities with higher community immigration rates, which is also supported by NST validation. These results indicate that the stochastic process plays a dominant role in shaping cropland bacterial community assembly, whereas the deterministic process plays a dominant role in shaping grassland bacterial community assembly.

## 5. Conclusions

The agro-pastoral ecotone of Tibet plays an important role in the safety of the ecological environment and national economic development for locals. Understanding the response of soil nutrients and microbial community composition to land-use change in cropland and grassland is of great significance to land management in the agro-pastoral ecotone of Tibet. We constructed a conceptual paradigm to reveal differences in soil nutrients and bacterial communities between cropland and grassland (Figure 7). The diversity and composition of grassland bacteria were more susceptible to environmental factors than those in cropland, which may lead the deterministic process to play a dominant role in shaping grassland bacterial community assembly. In addition, proper tillage may promote the connectivity and complexity of soil bacterial networks. Moreover, proper tillage also promotes bacterial migration and co-existence in an agro-pastoral ecotone. These results provide a new perspective for understanding the assembly mechanism of soil bacterial communities in the agro-pastoral ecotone.

## Figures and Tables

**Figure 1 microorganisms-10-01206-f001:**
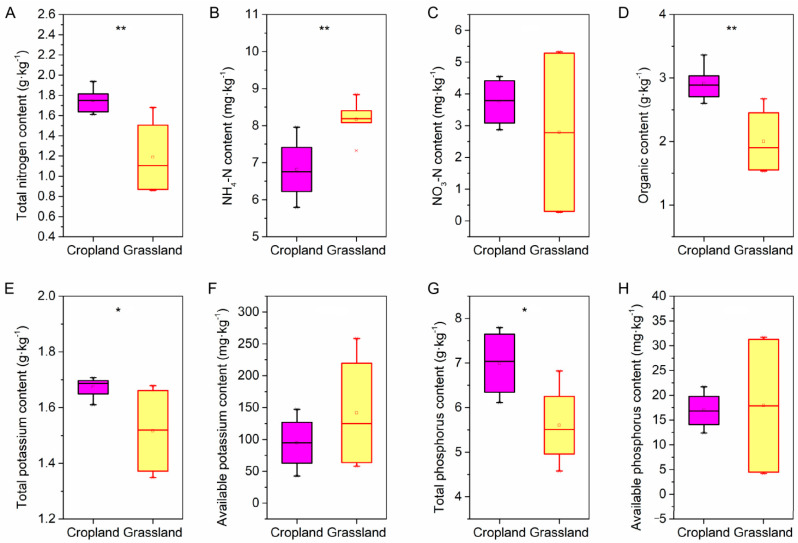
The difference in soil nutrients between cropland and grassland via independent-samples *t*-test. (**A**) The difference in soil total nitrogen between the cropland and grassland. (**B**) The difference in soil ammonia nitrogen (NH_4_-N) between the cropland and grassland. (**C**) The difference in soil nitrate-nitrogen (NO_3_-N) between the cropland and grassland. (**D**) The difference in soil organic matter between the cropland and grassland. (**E**) The difference in soil total potassium between the cropland and grassland. (**F**) The difference in soil available potassium between the cropland and grassland. (**G**) The difference in soil total phosphorus between the cropland and grassland. (**H**) The difference in soil available phosphorus between the cropland and grassland. ** shows the significant difference at the 0.01 level, and * shows the significant difference at the 0.05 level.

**Figure 2 microorganisms-10-01206-f002:**
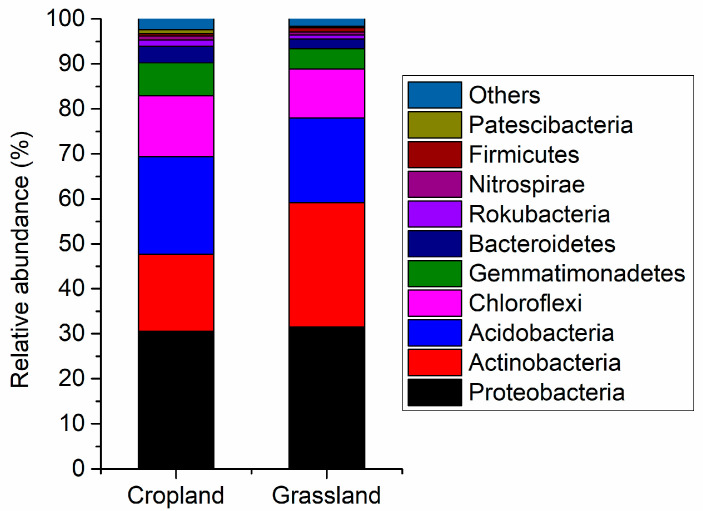
Relative abundance of main bacterial composition in the cropland and grassland soils at the phylum level.

**Figure 3 microorganisms-10-01206-f003:**
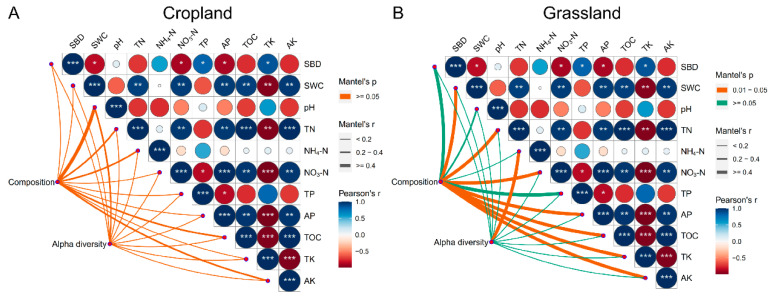
Mantel test of environmental variables with their corresponding composition and alpha diversity of bacterial communities in cropland (**A**) and grassland soils (**B**). The list of abbreviations is as follows: soil bulk density (SBD), soil water content (SWC), pH, total nitrogen (TN), ammonia nitrogen (NH_4_-N), nitrate-nitrogen (NO_3_-N), total phosphorus (TP), available phosphorus (AP), total potassium (TK), available potassium (AK), and organic matter (OM). *** shows the significant difference at the 0.01 level, ** shows the significant difference at the 0.01 level, and * shows the significant difference at the 0.05 level.

**Figure 4 microorganisms-10-01206-f004:**
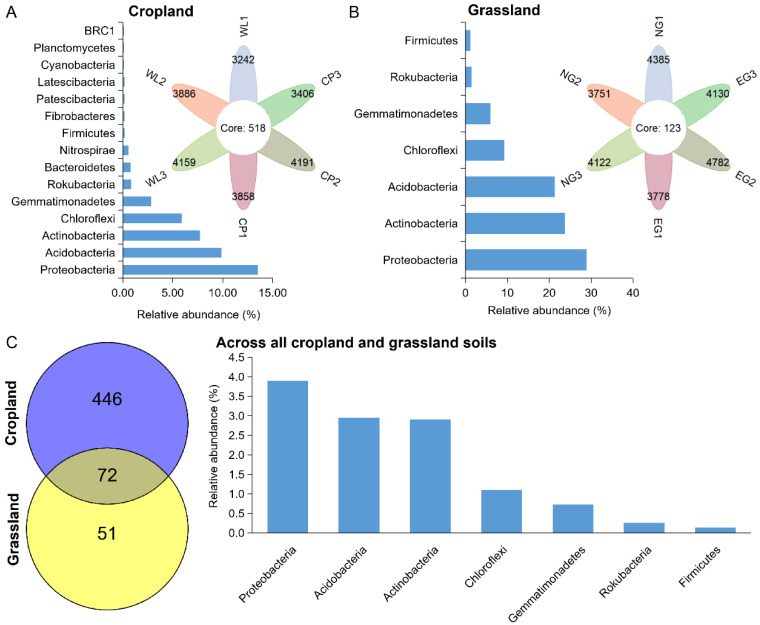
The composition of persistent bacteria in cropland (**A**) and grassland soils (**B**), and across all cropland and grassland soils (**C**).

**Figure 5 microorganisms-10-01206-f005:**
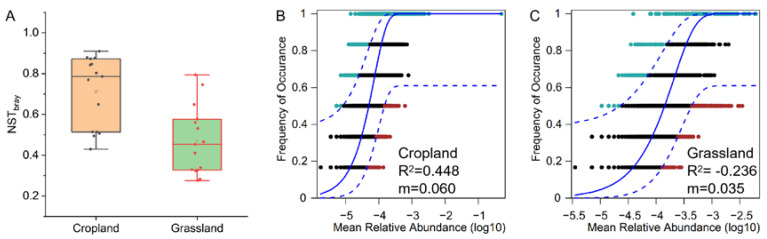
Assembly processes of bacterial communities in cropland and grassland soil from an agro-pastoral ecotone of Tibet. (**A**) Normalized stochasticity ratio (NST) of bacterial communities in cropland and grassland. A neutral community model was used to determine the contribution of stochastic processes to microbial community assembly in cropland (**B**) and grassland (**C**). In the model, “m” is the estimated migration rate, and the higher the value of “m”, the lower the degree of diffusion limitation of the bacterial community.

**Figure 6 microorganisms-10-01206-f006:**
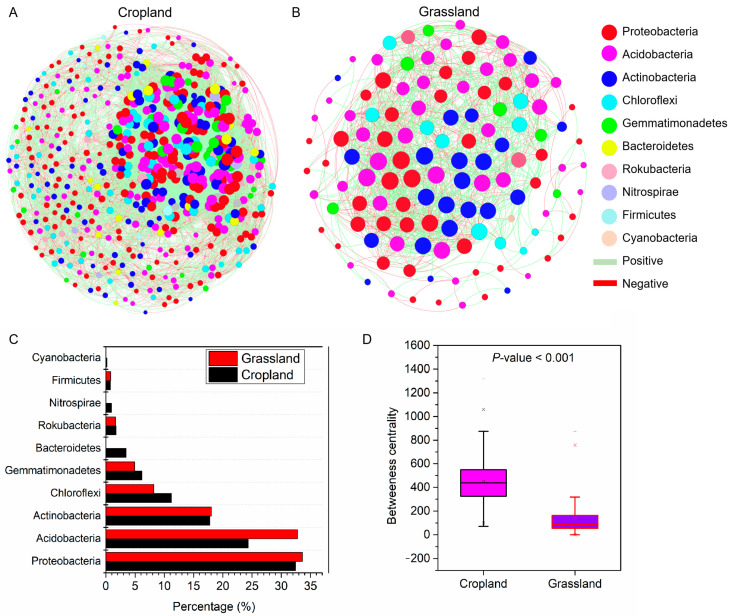
Co-occurrence networks of bacterial communities at the OUT level in the soil of cropland (**A**) and grassland (**B**) based on a Spearman correlation (|r| > 0.7, *p* < 0.05). The percentage of dominant bacteria in the networks (**C**). The difference in betweenness centrality of the network between cropland and grassland (**D**).

**Figure 7 microorganisms-10-01206-f007:**
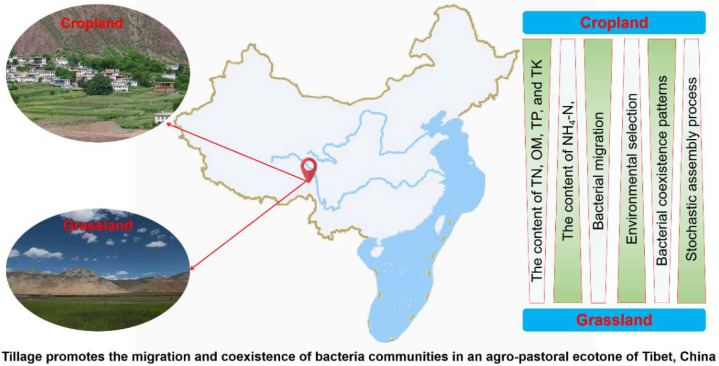
The conceptual model reveals the differences in the composition and assembly patterns of soil nutrients and bacterial communities in cropland and grassland. Green shows a decreasing trend from dark to light.

**Table 1 microorganisms-10-01206-t001:** Key topological features of co-occurrence networks of core bacterial taxa in the soil of cropland and grassland.

	Cropland	Grassland
Number of nodes	528	122
Number of edges	12,518	1153
Positive (%)	51.25	56.72
Negative (%)	48.75	43.28
Clustering coefficient	0.414	0.398
Graph density	0.093	0.156
Average connectivity	48.332	18.902
Average weighted degree	1.054	2.209

## Data Availability

Not applicable.

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
