# Peer review of "Tillage Promotes the Migration and Coexistence of Bacteria Communities from an Agro-Pastoral Ecotone of Tibet"

_microorganisms, 2022, doi:10.3390/microorganisms10061206_

Round 1
Reviewer 1 Report
The manuscript "Tillage promotes the migration and coexistence of bacteria communities from an agro‐pastoral ecotone of Tibet" cover a very significant subject.
The statistical approach and presentation are very nice
BUT the authors use the term "TILLAGE" and there are not information about (tillage system where used as Minimum or conventional tillage , equipment as chisel or plow etc)
Authors must to give more emphasis in tillage system!!!!
Author Response
Thank you for giving us the chance to revise the manuscript.
We cherish this chance to revise the manuscript as we can, according to the reviewers’ comments. We are very grateful that the reviewers’ comments are very constructive and helpful for improving the manuscript.
In the agro-pastoral ecotone of Tibet, the traditional ox-pulled plow is used as the main tillage system.
In order to take this reviewer's concern into account, and improve the quality of our manuscript, the introduction and methods were redescribed in the revised version of the manuscript, and corrections are highlighted in yellow color.
We hope that you find we have responded to the comments adequately, and the revised manuscript could be acceptable for publication in the journal.
We are looking forward to hearing from you soon.
Sincerely,
Weihong Zhang

Reviewer 2 Report
I read the reviewed article with curiosity. The microbiological composition of soil largely determines its quality (environmental values). The research results presented in the article show that the conversion of land - from grassland to cropland, causes major changes in the soil environment. Basically, the content of total nitrogen, organic material, total potassium and total phosphorus in soil changes - higher content on cropland. As a result of this process, the composition of soil bacteria changes. Importantly, tillage favors the unification of bacterial communities on arable soils. This suggests that the farmer / grower, by deciding to change the way he uses the land, can "manage the development of the soil environment". This may be of particular importance in the case of a shortage of arable land in various parts and regions of the world and the use of existing pastoral lands on cropland. On the newly created croplands, there is a greater migration of soil bacteria, and at the same time the accumulation and coexistence of bacterial communities, and consequently, the unification of these communities. This is, in my opinion, the most important conclusion from this article.
The cited dependence is perhaps not exceptionally revealing (it was possible to assume such a hypothesis), but it nevertheless confirms the scientifically obtained dependencies.
I have basically no comments on the reviewed article. Its individual parts (chapters) are correctly written. You can possibly correct the work in terms of the English language.
I recommend publishing this manuscript in Microorganisms.
26.05.2022.
Author Response
We are very grateful to the reviewers for their positive comments on our work
Round 2
Reviewer 1 Report
The authors add more details about tillage systems
Author Response

(The authors gave the same response as above.)
